# Socio-Demographic Composition and Potential Occupational Exposure to SARS-CoV2 under Routine Working Conditions among Key Workers in France

**DOI:** 10.3390/ijerph19137741

**Published:** 2022-06-24

**Authors:** Narges Ghoroubi, Emilie Counil, Myriam Khlat

**Affiliations:** 1The French Institute for Demographic Studies (Ined), 9 Cours des Humanités, 93322 Aubervilliers, France; emilie.counil@ined.fr (E.C.); khlat@ined.fr (M.K.); 2Doctoral School of Public Health (EDSP), Paris Saclay University, 16 Avenue Paul Vaillant Couturier, 94807 Villejuif, France; 3Institut de Recherche Interdisciplinaire sur les Enjeux Sociaux—Sciences Sociales, Politique, Santé, IRIS (UMR 8156 CNRS—EHESS—U997 INSERM), 5 Cours des Humanités, 93322 Aubervilliers, France

**Keywords:** COVID-19, essential workers, health equity, lockdown, socio-occupational disparities, worker protections, occupational health

## Abstract

This study aims to describe the socio-demographic profile of so-called “key workers” during the first lockdown in France and to assess their potential occupational exposure to SARS-CoV-2 under routine, pre-pandemic working conditions. We used the French list of essential jobs that was issued during the first lockdown to identify three subgroups of key workers (hospital healthcare, non-hospital healthcare, non-healthcare). Based on the population-based “Conditions de travail-2019” survey, we described the socio-demographic composition of key workers and their potential work-related exposures (to “infectious agents,” “face-to-face contact with the public,” and “working with colleagues”) using modified Poisson regression. In general, women, clerical and manual workers, workers on temporary contracts, those with lower education and income, and non-European immigrants were more likely to be key workers, who accounted for 22% of the active population. Non-healthcare essential workers (57%) were the most socially disadvantaged, while non-hospital healthcare workers (19%) were polarized at both extremes of the social scale; hospital healthcare workers (24%) were intermediate. Compared to non-key workers, all subgroups had greater exposure to infectious agents and more physical contact with the public. This study provides evidence of accumulated disadvantages among key workers concerning their social background, geographical origin, and potential SARS-CoV-2 exposure.

## 1. Introduction

The COVID-19 crisis brought the occupational hazard of contamination in the workplace to the fore as a large-scale public health issue [1]. The common view was that risks were linked to direct contact with people who were infected by COVID-19. Therefore, the occupations that were considered to be “on the front line” during the very first months of the epidemic were health professionals, and more specifically, hospital caregivers [2,3]. However, these occupations (physicians, nurses, care assistants) were concerned to a different extent depending on their actual working conditions. In addition, other less visible jobs were in contact with the virus through infected materials (cleaning staff, funeral workers) or brought into frequent contact with colleagues or the public, while some commuted to work through public transportation.

State restrictions in response to the first wave of COVID-19 rapidly brought worldwide attention to the broader concept of “essential workers”. They were previously defined as workers who provide services, “the interruption of which would endanger the life, personal safety or health of the whole or part of the population”, according to the International Labor Organization (ILO) [4]. From the onset of the pandemic, many governments scrambled to identify the services and workers that they deemed to be essential, with varying terminologies [5]. In the United States, for example, all workers whose job served to “protect their communities, while ensuring continuity of functions critical to public health and safety, as well as economic and national security” [6] were designated as essential workers, even if they could work from home, and a narrower subgroup who had to provide their labor in person were considered as “frontline essential workers” [7].

In the US, pre-pandemic surveys showed that many of these potentially high-risk frontline jobs were held by individuals at the most disadvantaged intersections of gender, racial/ethnic minority status, and socio-occupational class [8]. Disparities in working arrangements (e.g., telework, onsite work, layoffs, and sick leave), as well as disparities in working conditions among onsite workers (e.g., regarding exposure to the virus), have therefore been hypothesized as a driving part of COVID-19 disparities among working-age adults. It has also been pointed out that a substantial proportion of those workers were often already at a heightened risk for poor health due to multiple systemic social and economic disadvantages that existed pre-pandemic [9]. These early observations are in line with the now-common view that COVID-19 is a “syndemic pandemic” embedded in inequalities in chronic conditions, such as existing non-communicable diseases, as well as inequalities in the social determinants of health [10,11]. Due to both the syndemic nature of the pandemic and the diversity of working arrangements and workplace exposures, occupational disparities in COVID-19 infection and outcomes are expected to be non-linear—in other words, not to follow a gradient along the socio-occupational hierarchy.

Whereas studies have found that working-class individuals may have been less protected by the lockdown measures than more privileged categories due to their professional obligations, their living conditions in overcrowded housing and densely populated areas, and their pre-existing comorbid health conditions [12], the lower social class has also been shown to be at a greater risk of income and job loss, which correlates with reduced workplace exposure [13,14]. Moreover, the impact of the lockdown on the middle class may have been mixed, as it may have protected those who could benefit from work-at-home arrangements or government-subsidized layoffs, while healthcare workers have been disproportionately exposed [15]. The extent to which work contributed to the social gradient of infection also depends on the policy context, in particular the stringency of the workplace measures (workplace closures and onsite health protection measures) that varied between and within countries and over time.

In France, the government imposed a strict 2-months lockdown on 17 March 2020, which was rated as one of the most stringent “lockdown style policies” by the Oxford COVID-19 Government Response Tracker [16]. In this country, “key workers” were defined merely as workers who continued to work on-site during the first lockdown [17]. Many industries were shut down or faced a sharp decline in demand and remote work was imposed on the majority of office workers. Even so, several million workers were designated as “key workers” who worked in the healthcare sector, as well as other services, such as food and agriculture (food workers, cashiers); sanitation (water treatment, waste collection, cleaning, funeral services); logistics (transportation, warehousing); or utilities (energy providers: electricity, oil, gas, computer services, construction) [17]. Working conditions are already known to be important social determinants of health [18]; in the absence of many other usual social interactions, they may have played an even more important role in explaining the disproportionate burden of COVID-19 among disadvantaged workers and within their households, but also, for example, among middle-class social and health workers.

While we know already that the racial division of work translated into a higher probability of working in jobs that were deemed to be essential among immigrant and minority populations [19], little is known in France about the diversity of sociodemographic profiles among workers who continued to work in person during the first lockdown and their work-related exposure to SARS-CoV-2.

Our study builds upon pre-COVID-19 employment survey data to provide insight into the social disparities between different subgroups of ‘key workers’, by describing their socio-demographic profile and assessing their potential occupational exposure to SARS-CoV-2, and by extension, to infectious diseases of respiratory origin. We hypothesize that minimal distinctions should be made, notably between hospital-healthcare workers, non-hospital healthcare workers, and other key workers outside of healthcare to better understand the social gradient of COVID-19 that was observed in the early pandemic at the intersection of gender, class, and race/ethnicity [12,15,20].

## 2. Materials and Methods

### 2.1. Data

We used the 2019 edition of the “Conditions de travail” survey (CT-2019) which is a periodic population-based cross-sectional study conducted by the French Ministry of Labor on a nationally representative sample of 24,951 working adults representing 27 million employees in the public and private sectors, as well as the self-employed [21]. We selected participants aged 18 to 64 years who were living in Metropolitan France (N = 23,231).

### 2.2. Socio-Demographic Variables

In addition to gender and age, we considered the following socio-demographic variables. Socio-occupational group was defined following the French classification of occupations and socio-occupational categories and grouped into farmers; self-employed (excluding intellectual occupations); senior executive professionals and professors; middle executive professionals; clerical workers; and manual workers. Clerical and manual workers were further split into skilled and low-skilled subgroups. The employment status variable distinguished private and public sector workers. The type of contract had four categories: apprenticeship, independent, permanent, and temporary contract. Educational level was converted to the corresponding International Standard Classification of Education (ISCED) [22]. Income level was defined based on declared monthly income deciles. Finally, geographic origin divided workers into: persons who were born in the French overseas departments (DOM); descendants of immigrants, including those who were born in Metropolitan France with at least one parent born outside Metropolitan France without having French nationality at birth; immigrants who were born outside Metropolitan France without having French nationality at birth; and the mainstream population, which consisted of other persons residing in Metropolitan France [23]. From there, we built a 14-level origin variable for which immigrants and the descendants of immigrants were divided into six geographic regions: Maghreb (Algeria, Morocco, Tunisia); all other African countries; Asia; European Union (15 countries); all other EU countries; and other countries.

### 2.3. Key Worker’s Definition

We used a list of 35 occupations belonging to the health, food, public services, and logistics sectors, to be considered “key occupations” during the first lockdown in France [24]. The list does not include teaching, childcare, or construction workers, as those remained mostly locked down during the first wave of the pandemic. The key occupations were further categorized into three sub-groups: hospital healthcare workers, non-hospital healthcare workers, and non-healthcare essential workers (Appendix A
Table A1).

Due to the aggregate level of information in the CT-2019 database on some occupations, such as funeral staff, salespeople, and public transport drivers, some of these groups contained workers who were likely not working onsite during the first lockdown. We performed a sensitivity analysis by removing these occupations to explore whether there was any change in the socio-demographic profile of key workers and their work-related exposure to SARS-CoV-2.

### 2.4. Work-Related Exposure to SARS-CoV-2

To estimate the percentage of potentially exposed workers, we used information on the routine working conditions prevailing right before the COVID-19 pandemic: for infectious agents: “At your workplace, are you exposed to infectious risks”?; for face to face contact with the public: “Are you in direct contact with the public? (users, patients, students, travelers, customers, suppliers,…)”, and then, in the case of a positive reply, “is the contact face to face”?; for physical contact with colleagues: “Do you work alone?“. All of the exposure factors were coded as binary variables.

### 2.5. Modeling

To describe the sociodemographic profile of key workers, we first considered as independent variables the sociodemographic characteristics and their interactions, and as outcomes, being a key worker and belonging to a key worker sub-group. To model the probability of occupational exposure to SARS-CoV-2, we then selected being a key worker as the independent variable, and the outcomes were in turn the three occupational exposure factors.

In both models, the outcome of interest was fairly common (i.e., more than 10%) among the sociodemographic and occupational subgroups. As the log-binomial model did not converge in some cases, we performed a modified Poisson regression to estimate the prevalence ratio (PR) of being a key worker in the first model and being exposed to workplace exposure factors in the second model [25]. When possible, we ran both models and compared the results. Almost all estimates of PR were similar to one decimal place and, as anticipated, the confidence intervals were larger in the modified Poisson regression.

We also investigated the association between the intersections of gender, geographic origin, and occupational group with being a key worker through an 8-level variable that divided participants based on being male or not, born in Metropolitan France or not, and being a senior executive professional or not.

The data were weighted to be nationally representative of the working population in Metropolitan France. An extensive description of the sampling and weighting methods that are used in CT periodic surveys is available elsewhere [26]. Both the descriptive analysis and modeling were carried out using SAS software (Linux distribution), version 9.4 on SAS Studio interface, version 3.6, released by SAS Institute Inc., Cary, NC, USA.

## 3. Results

At the start of the pandemic, key workers accounted for 22% of workers in Metropolitan France. For this analysis, we divided those key workers into hospital healthcare workers (24%), non-hospital healthcare workers (19%), and non-healthcare essential workers (57%). The distributions of the main sociodemographic characteristics of key workers (taken as a whole and in different subgroups) are shown in Table 1, and the stratified multivariate modeling carried for each subgroup are shown in Table 2.

Compared to non-key workers, key workers as a whole were more likely to be female, of younger age, and born in the Maghreb. On the opposite, there was a negative association with being born in Asia, and, to an even greater extent, with upper occupational and educational levels (Table 1 and Table 2).

Figure 1 shows that gender and occupational group had significant interactive effects on the likelihood of being a key worker, with female non-executive professionals having the highest probability, compared to male senior executive professionals who were born in France. However, being born in Metropolitan France did not meaningfully change the likelihood of being a key worker in an intersection of gender, geographic origin, and occupational group.

Looking at different subgroups of key workers, we found more contrasted profiles. Among hospital healthcare workers, there was a higher percentage of females than among non-key workers, as well as higher percentages of the 25–34 age group, middle executive professionals, skilled and low-skilled clerical workers, public sector workers, those on temporary contracts, workers with upper secondary or short-cycle tertiary education or a bachelor’s degree, those in the intermediate (fourth–seventh) income decile, and those from overseas France (Table 1 and Table 2).

Among non-hospital healthcare workers, there was also a higher percentage of women than among non-key workers, as well as higher percentages of older participants, low-skilled clerical workers, private-sector workers, independent workers, those with a master’s degree or higher, individuals in the extreme (highest and lowest) income deciles, and African immigrants. This group of key workers appeared to include workers from both ends of the social scale, for example, home carers and private-sector physicians (Table 1 and Table 2).

Among non-healthcare essential workers, there was no gender imbalance. However, there was a higher percentage in this group than among non-key workers of those under 34 years of age and private-sector workers. All occupational categories, particularly clerical and manual workers, were more likely to be in the third group of essential workers, compared with senior executives. Concerning education and income level, working in this group exhibited a pronounced social gradient, being more likely among the least educated and poorest. Those with apprenticeship or self-employment contracts and the descendants of European immigrants were less likely to hold these jobs, while Asian descendants and Maghrebi immigrants were disproportionately employed in these occupations (Table 1 and Table 2).

Modified Poisson regression modeling of the probability of being exposed to infectious agents, having face-to-face contact with the public, and working with coworkers indicated that, compared with non-key workers, the key workers in each subgroup were more likely to be exposed to infectious agents and to have physical contact with the public under routine working conditions. However, working with colleagues differed by key worker subgroup, as the hospital healthcare workers worked more often with co-workers, and non-hospital healthcare workers and non-healthcare essential workers worked more often on their own. The first group of key workers was particularly exposed to all three risk factors (Table 3).

As explained in the methods section, a sensitivity analysis was performed by excluding funeral staff (N = 4), salespeople (N = 582), and public transportation drivers (N = 62) from the non-healthcare essential workers sub-group due to a coding inaccuracy. This exclusion did not substantially alter the sociodemographic profile or exposure to the three exposure factors among them, except that males became more represented in this key worker subgroup compared with females (Appendix A
Table A2 and Table A3).

## 4. Discussion

In the spring of 2020, about 22% of the active population in France could not be locked down because of the essential nature of their work activity, according to the data that was collected before the pandemic and based on the definition of key workers in use in the country. This is in line with other estimates of the share of key workers during the first lockdown in France [19,24], and close to the reported proportion in Italy (25%) [27]. Higher figures were reported in the US (43%) [7], in the UK (33%) [28], and in Europe as a whole (31%) [29]. This difference in the share of key workers may result from the stringency of the first lockdown in France, which is also the reason why we find that the key health worker and the key non-healthcare essential worker represent an almost equal share of key workers, whereas in the US, for example, healthcare workers represent 20% of frontline workers [7].

The National Bureau of Economic Research described frontline workers in the US as, on average, a less educated group, lower-paid than all workers, with a higher share of men, and including more racial and ethnic minorities (particularly Hispanics) and immigrants [7]. The Center for Economic and Policy Research provided a similar demographic profile of key workers in the US, except for an over-representation of women in essential services, especially healthcare, child care, and social services [30]. In a French study, women, immigrants, and people who were born in French overseas departments, as well as individuals with lower labor market protection (part-time, no contract, lower-paid) were found to be more likely to work in a frontline job [19]. In Europe as a whole, non-European immigrants were found to be overrepresented among key workers, particularly in low-skilled key occupations (e.g., personal care workers in health service, transport and storage laborers, drivers, food processing workers) [29]. Similarly, in the US, people of color, Black, and Hispanic/Latino workers in particular were at a high risk of being employed in key industries [31,32].

In line with the literature, we noted that although the set of key occupations was widely diverse, ranging from highly skilled jobs such as physicians to manual jobs such as drivers and construction workers, key workers had on average a disproportionately lower-income and were less educated. We also found that women were overrepresented in healthcare key occupations. Although many non-healthcare essential jobs are gendered (some predominantly male, such as public transportation drivers, firefighters, and police forces, and some predominantly female, such as cashiers and cleaners [33]), we found no gender differences among overall non-healthcare essential workers. Regarding origin, our results showed a specific pattern: DOM natives were overrepresented among hospital healthcare workers; African immigrants (in particular, Maghrebi) among the healthcare workers outside the hospital setting; and Maghrebi immigrants, as well as descendants of Asian immigrants among non-healthcare essential workers.

Given that healthcare workers are at a particularly high risk of exposure to infectious agents due to the nature of their patient care work, their occupational health and safety have often been highlighted in the literature and various media. One systematic review synthesized the major work-related risk factors for COVID-19 among healthcare workers as: exposure to SARS-CoV-2 (caring for COVID-19-positive patients, working in high-prevalence regions); lack of personal protective equipment (PPE) (inadequate PPE, re-used PPE, unqualified handwashing); and workplace setting (inpatient settings, nursing homes, sharing the work environment with co-workers, longer working hours) [3].

On the other hand, little has been published on workplace risk assessment for non-healthcare essential workers. To address this knowledge gap, Gaitens et al. conducted a narrative review of the peer-reviewed and gray literature, as well as news sources. They summarized the work-related COVID-19 risks for nonmedical key workers as follows: inability to respect physical distancing (working on long production lines in close proximity to co-workers, encountering a high volume of customers or public, who may or may not be wearing masks, especially in the early days of the pandemic); limited availability of PPE and other safety supplies (poor hygiene, lack of training on health protocols); workplace characteristics (small, crowded, unventilated, cold and damp spaces); limited labor rights (lack of sick leave, incentives that may encourage workers to work while ill); and poor testing and contact tracing strategies (COVID-19 test shortage, non-reporting of exposure to infected co-workers) [5]. Further to these adverse working conditions, we could hypothesize that work pressure may lead to less compliance with prevention measures in these workplaces.

Based on a 2019 national survey and on the three work-related exposure factors that we identified in this study, we obtained similar results showing that key workers were likely at an increased risk of exposure to SARS-CoV-2 because they had a higher risk of exposure to infectious agents and face-to-face contact with the public when compared to non-key workers. Those who were most exposed to these health hazards were, as expected, the healthcare workers.

The key strength of this study lies in the fact that the CT-2019 survey is a large, established national survey that provides information on the working conditions of the entire labor force in France. This allowed us to study detailed occupational categories beyond the health sector and compare different subgroups of key workers with non-key workers. Another asset of the study is that the data were collected in the year before the onset of the pandemic and hence not long before the first lockdown. Given that the relevant working conditions of those who continued to work on-site during the first lockdown did not change much from the usual working conditions, the CT-2019 survey could be an appropriate source to study this period. Finally, in this study, we took an intersectional approach to investigate socio-occupational disparities in key jobs across genders and immigrants, compared to the general population.

Our study also has some limitations, the main one being the quantification of “exposure to infectious agents” and “working with co-workers”. The question on exposure to infectious agents was not intended to measure specifically viral contamination or even exposure to a human reservoir of infectious agents, so some jobs that were in contact with other sources of infectious agents, such as animal reservoirs, were also classified as exposed (e.g., veterinarian). This may have overestimated the exposure prevalence in all the sub-groups of key workers. We used “not working alone” as the closest proxy for physical contact with co-workers. However, working alone in the context of a survey that is not designed to study COVID-19 exposures could imply working autonomously and without collaboration with co-workers, rather than no physical contact with them. Thus, some workers who had face-to-face contact with co-workers might have reported no contact since they work independently, and conversely, some workers with no physical contact with colleagues might have reported working with colleagues since they work in a team setting to complete certain tasks. Finally, the CT surveys failed to include informal and undocumented workers who were likely to be more vulnerable to adverse working conditions and lack of adequate PPE if they continued to work and had no access to the social protection measures that were implemented during the pandemic, such as unemployment benefits in case of job loss [34]. It is unlikely that there were undocumented healthcare workers in the hospitals, however, we believe that there may have been undocumented non-hospital healthcare workers, and probably even more undocumented workers in non-healthcare essential services who continued to work on-site during the first lockdown in France.

Further studies are needed to capture the real-time dynamics of work-related SARS-CoV-2 exposure among different socio-occupational categories throughout the COVID-19 pandemic. Particularly, other exposure mechanisms, such as housing conditions, commuting modes and duration, and living in a densely populated area need to be taken into account to capture the mechanisms by which social inequalities unfolded [20]. Other groups of workers, such as educational and social work staff who were added to the list of key workers after the first lockdown and who have been shown to be at a high risk of exposure to SARS-CoV-2 in their work should also be considered in the further analysis [35,36]. This could ultimately lead to an estimation of the contribution of working as a key worker in the disproportionate COVID-19 infection that is found among healthcare workers and the working classes [37,38].

## 5. Conclusions

In this study, we provide a detailed description of the sociodemographic profile of the different subgroups of key workers in the French population. Compared to other workers, key workers in all subgroups have greater exposure to infectious agents and more physical contact with others. Of all subgroups, the non-healthcare essential workers were the most socially disadvantaged, while non-hospital healthcare workers were the most socially polarized and the least exposed to working with colleagues.

Although we observe a diversity of social backgrounds among key workers and heterogeneity in their potential work-related exposure to SARS-CoV-2, individuals from lower sociodemographic class are, on average, more likely to be exposed to SARS-CoV-2 by continuing to perform their jobs in person—jobs that most often put them into increased close contact with infectious agents and the public, even under routine working conditions. There is an urgent need to effectively protect these key workers and to ensure strict occupational health surveillance in their workplaces.

## Figures and Tables

**Figure 1 ijerph-19-07741-f001:**
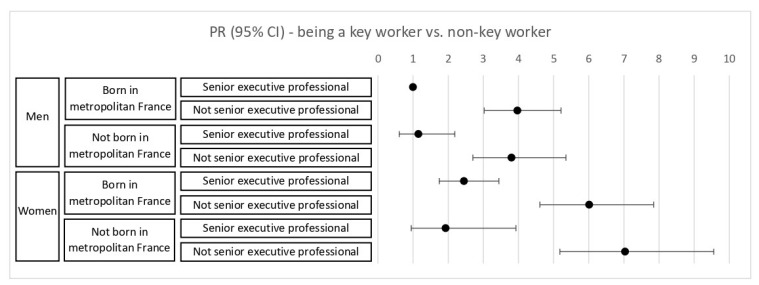
Estimated age-adjusted prevalence ratios (PR) for the association between intersecting socio-demographic characteristics (gender, origin, occupation) and working as a key worker.

**Table 1 ijerph-19-07741-t001:** Sociodemographic characteristics of the general active population, non-key workers, the three sub-groups of key workers, and all key workers.

	Non-Key Workers	Hospital HCW ^a^	Non-Hospital HCW ^a^	Essential Non HCW ^a^	All Key Workers ^b^	All Active Population ^c^
	N (%)	N (%)	N (%)	N (%)	N (%)	N (%)
**Total**	17,156 (78.6)	2738 (5.2)	1009 (4.1)	2291 (12.1)	6038 (21.5)	23,194 (100)
**Gender**						
Male	8371 (55.1)	403 (17.2)	153 (15.5)	1286 (56.1)	1842 (38.9)	10,228 (51.5)
Female	8785 (45.0)	2335 (82.8)	856 (84.5)	1005 (43.9)	4196 (61.1)	13,003 (48.5)
**Age group**						
18–24	455 (7.1)	78 (6.5)	21 (7.0)	107 (11.1)	206 (9.2)	665 (7.5)
25–34	2971 (22.6)	651 (27.4)	177 (17.1)	433 (19.9)	1261 (21.2)	4236 (22.2)
35–44	4245 (25.2)	624 (24.6)	225 (22.1)	582 (23.4)	1431 (23.5)	5684 (24.8)
45–54	5840 (28.4)	883 (26.6)	324 (29.8)	758 (27.8)	1965 (27.9)	7816 (28.3)
55–64	3645 (16.8)	502 (15.0)	262 (24.1)	411 (17.8)	1175 (18.3)	4830 (17.2)
**Occupational group**						
Farmers	432 (1.8)	0 (0.0)	0 (0.0)	0 (0.0)	0 (0.0)	432 (1.4)
Self-employed	947 (7.0)	0 (0.0)	0 (0.0)	104 (5.3)	104 (3.0)	1051 (6.1)
Senior executive professionals and Professors	3838 (21.6)	251 (10.1)	152 (18.1)	24 (0.9)	427 (6.5)	4265 (18.3)
Middle executive professionals	5236 (28.0)	1262 (38.2)	440 (29.0)	279 (7.9)	1981 (19.3)	7217 (26.1)
Skilled clerical workers	2352 (12.3)	827 (30.1)	0 (0.0)	387 (11.4)	1214 (13.8)	3566 (12.6)
Skilled manual workers	1881 (12.6)	0 (0.0)	0 (0.0)	644 (30.8)	644 (17.4)	2525 (13.6)
Low-skilled clerical workers	1647 (10.1)	398 (21.5)	417 (53.0)	547 (28.2)	1362 (31.4)	3009 (14.7)
Low-skilled manual workers	821 (6.5)	0 (0.0)	0 (0.0)	306 (15.4)	306 (8.7)	1127 (7.0)
**Employment status**						
Private	11,118 (80.0)	858 (37.5)	763 (87.6)	1791 (90.8)	3412 (77.3)	14,555 (79.4)
Public	6038 (20.0)	1880 (62.5)	246 (12.4)	500 (9.2)	2626 (22.8)	8676 (20.6)
**Type of contract**						
Apprenticeship	405 (5.3)	38 (2.6)	22 (3.1)	59 (4.2)	119 (3.6)	525 (4.9)
Independent	1749 (10.7)	8 (0.2)	312 (26.8)	122 (5.8)	442 (8.5)	2195 (10.3)
Permanent	14,052 (74.4)	2490 (79.2)	618 (60.1)	1974 (78.7)	5082 (75.2)	19,160 (74.5)
Temporary	950 (9.6)	202 (18.0)	57 (10.0)	136 (11.3)	395 (12.7)	1351 (10.3)
**Education level**						
No formal education	835 (5.7)	84 (4.8)	60 (6.4)	254 (12.9)	398 (9.7)	1233 (6.6)
Primary or lower secondary education	802 (5.1)	96 (3.9)	56 (8.3)	205 (9.4)	357 (7.8)	1160 (5.7)
Upper secondary education	3883 (22.4)	601 (24.0)	199 (24.3)	817 (37.3)	1617 (31.6)	5508 (24.4)
Post-secondary non-tertiary education	3296 (19.1)	372 (15.3)	101 (13.3)	530 (22.4)	1003 (18.9)	4307 (19.1)
Short-cycle tertiary education or bachelor’s or equivalent	5459 (30.3)	1242 (38.3)	365 (24.7)	429 (15.7)	2036 (22.9)	7511 (28.8)
Master’s or equivalent or higher	2860 (17.4)	337 (13.8)	226 (23.1)	54 (2.3)	617 (9.1)	3481 (15.6)
**Income level**						
Decile 1 (lowest)	1649 (13.6)	118 (11.0)	211 (31.7)	271 (15.3)	600 (17.3)	2256 (14.5)
Decile 2&3	3104 (20.3)	513 (24.0)	221 (27.0)	690 (35.6)	1424 (31.1)	4535 (22.6)
Decile 4&5	3048 (17.4)	771 (27.2)	93 (6.0)	399 (17.9)	1263 (18.0)	4318 (17.6)
Decile 6&7	3395 (18.5)	718 (22.1)	88 (5.3)	468 (17.7)	1274 (16.4)	4674 (18.0)
Decile 8&9	3640 (19.6)	400 (10.5)	162 (12.3)	334 (11.0)	896 (11.1)	4541 (17.8)
Decile 10	1821 (10.6)	187 (5.2)	184 (17.6)	66 (2.6)	437 (6.1)	2262 (9.6)
**Geographical origin**						
Mainstream population	15,227 (85.2)	2472 (86.5)	897 (81.1)	2005 (84.6)	5374 (84.4)	20,632 (85.0)
DOM natives	170 (1.0)	46 (2.4)	7 (0.5)	27 (1.3)	80 (1.4)	250 (1.1)
Descendants of immigrant(s) from Maghreb	145 (1.4)	16 (0.9)	3 (0.7)	22 (1.5)	41 (1.2)	186 (1.3)
Descendants of immigrant(s) from all other African countries	33 (0.3)	3 (0.1)	1 (0.6)	6 (0.3)	10 (0.3)	43 (0.3)
Descendants of immigrant(s) from Asia	26 (0.3)	5 (0.2)	0 (0.0)	8 (1.0)	13 (0.6)	39 (0.4)
Descendants of immigrant(s) from European Union 15 countries ^d^	282 (2.1)	28 (0.9)	14 (3.3)	33 (1.4)	75 (1.6)	358 (2.5)
Descendants of immigrant(s) from all other European countries	26 (0.2)	3 (0.3)	2 (0.2)	6 (0.2)	11 (0.2)	37 (0.2)
Descendants of immigrant(s) from other countries	17 (0.2)	4 (0.2)	1 (0.1)	0 (0.0)	5 (0.1)	22 (0.2)
Immigrants from Maghreb	262 (2.2)	30 (2.2)	16 (4.6)	56 (3.7)	102 (3.5)	366 (2.5)
Immigrants from all other African countries	198 (1.8)	53 (2.3)	19 (3.9)	28 (1.8)	100 (2.3)	298 (1.9)
Immigrants from Asia	155 (1.6)	8 (0.1)	6 (0.7)	24 (1.1)	38 (0.8)	193 (1.4)
Immigrants from European Union 15 countries ^d^	293 (2.2)	25 (1.0)	23 (2.4)	33 (1.9)	81 (1.8)	375 (2.1)
Immigrants from all other European countries	91 (0.9)	13 (2.7)	4 (0.7)	8 (0.8)	25 (1.2)	117 (1.0)
Immigrants from other countries	74 (0.6)	12 (0.4)	7 (1.1)	9 (0.5)	28 (0.6)	103 (0.6)

^a^ HCW: Healthcare workers. ^b^ All key workers = “hospital HCW” + “non-hospital HCW” + “Essential non HCW”. ^c^ All active population = “Non-key workers” + “All key workers”. ^d^ The OECD definition of European Union 15 countries (EU15): https://stats.oecd.org/glossary/detail.asp?ID=6805 (accessed on 14 June 2022).

**Table 2 ijerph-19-07741-t002:** Estimated adjusted prevalence ratios (PR) for the association between socio-demographic characteristics and working as a key worker in each sub-group of key workers, compared to non-key workers (stratified multivariate models).

	Hospital HCW ^a^	Non-Hospital HCW ^a^	Essential Non HCW ^a^	All Key Workers
	Adjusted PR ^b^ (vs. Non-Key Workers)			
**Gender**				
Male	Ref	Ref	Ref	Ref
Female	5.4 ^c^ (4.4–6.7)	6.3 (4.8–8.2)	1.1 (1.0–1.2)	1.8 (1.6–1.9)
**Age group**				
18–24	1.2 (0.8–1.8)	0.9 (0.5–1.8)	1.6 (1.2–2.3)	1.3 (1.0–1.7)
25–34	1.5 (1.1–1.9)	0.6 (0.4–0.8)	1.3 (1.0–1.6)	1.1 (1.0–1.3)
35–44	1.1 (0.9–1.4)	0.7 (0.5–0.9)	1.2 (1.0–1.5)	1.0 (0.9–1.2)
45–54	1.1 (0.9–1.3)	0.8 (0.6–1.0)	1.0 (0.9–1.2)	1.0 (0.9–1.1)
55–64	Ref	Ref	Ref	Ref
**Occupational group**				
Farmers	-	-		
Self-employed	-	-	15.9 (8.8–28.8)	1.5 (1.1–2.0)
Senior executive professionals and Professors	Ref	Ref	Ref	Ref
Middle executive professionals	2.4 (1.9–3.1)	1.1 (0.8–1.5)	6.1 (3.4–10.8)	2.0 (1.7–2.4)
Skilled clerical workers	3.1 (2.4–3.9)	-	18.5 (10.5–32.9)	2.7 (2.2–3.2)
Skilled manual workers	-	-	42.1 (24.4–72.5)	4.2 (3.5–5.1)
Low-skilled clerical workers	3.1 (2.3–4.2)	3.4 (2.6–4.6)	47.1 (27.1–82.0)	5.4 (4.5–6.5)
Low-skilled manual workers	-	-	41.9 (24.0–73.2)	3.7 (3.0–4.7)
**Employment status**				
Private	Ref	Ref	Ref	Ref
Public	4.6 (3.8–5.6)	0.4 (0.3–0.6)	0.5 (0.4–0.6)	1.1 (1.0–1.2)
**Type of contract**				
Apprenticeship	0.6 (0.4–1.0)	0.8 (0.4–1.7)	0.5 (0.4–0.8)	0.6 (0.5–0.9)
Independent	0.0 (0.0–0.1)	3.4 (2.7–4.2)	0.6 (0.4–0.7)	0.9 (0.8–1.1)
Permanent	Ref	Ref	Ref	Ref
Temporary	1.5 (1.1–2.0)	1.1 (0.7–1.6)	0.9 (0.7–1.2)	1.0 (0.9–1.2)
**Education level**				
No formal education	1.2 (0.7–2.0)	0.7 (0.4–1.0)	14.0 (8.2–23.9)	2.6 (2.1–3.3)
Primary or lower secondary education	1.0 (0.7–1.6)	1.0 (0.6–1.6)	12.0 (7.0–20.8)	2.4 (1.9–3.0)
Upper secondary education	1.7 (1.2–2.3)	0.9 (0.6–1.3)	11.0 (6.5–18.5)	2.4 (2.0–2.9)
Post-secondary non-tertiary education	1.1 (0.8–1.5)	0.5 (0.3–0.8)	7.6 (4.4–13.0)	1.7 (1.4–2.0)
Short-cycle tertiary education or bachelor’s or equivalent	1.4 (1.0–1.9)	0.6 (0.4–0.8)	3.6 (2.1–6.2)	1.3 (1.1–1.6)
Master’s or equivalent or higher	Ref	Ref	Ref	Ref
**Income level**				
Decile 1 (lowest)	0.9 (0.6–1.4)	0.8 (0.6–1.0))	4.7 (3.1–7.1)	1.6 (1.4–2.0)
Decile 2&3	1.4 (1.0–1.9)	0.5 (0.4–0.7)	7.0 (4.8–10.3)	2.0 (1.6–2.4)
Decile 4&5	2.0 (1.4–2.6)	0.2 (0.1–0.3)	4.4 (2.9–6.5)	1.5 (1.3–1.8)
Decile 6&7	1.6 (1.2–2.2)	0.1 (0.1–0.2)	4.1 (2.8–6.1)	1.4 (1.2–1.7)
Decile 8&9	0.9 (0.6–1.2)	0.3 (0.2–0.5)	2.4 (1.6–3.7)	1.0 (0.8–1.2)
Decile 10	Ref	Ref	Ref	Ref
**Geographical origin**				
Mainstream population	Ref	Ref	Ref	Ref
DOM natives	2.1 (1.3–3.4)	0.5 (0.2–1.3)	1.2 (0.7–1.9)	1.2 (0.9–1.7)
Descendants of immigrant(s) from Maghreb	0.7 (0.4–1.3)	0.7 (0.2–2.2)	0.9 (0.5–1.8)	0.8 (0.5–1.3)
Descendants of immigrant(s) from all other African countries	0.3 (0.1–1.3)	1.6 (0.2–12.3)	0.9 (0.3–2.4)	0.9 (0.4–2.2)
Descendants of immigrant(s) from Asia	0.5 (0.1–1.6)	-	2.2 (1.1–4.7)	1.4 (0.7-.2.9)
Descendants of immigrant(s) from European Union 15 countries ^d^	0.4 (0.2–0.7)	1.5 (0.7–3.3)	0.6 (0.4–0.9)	0.7 (0.5–1.0)
Descendants of immigrant(s) from all other European countries	1.2 (0.4–3.9)	1.1 (0.3–4.0)	1.3 (0.6–2.9)	1.1 (0.7–2.0)
Descendants of immigrant(s) from other countries	0.9 (0.2–4.3)	0.6 (0.1–5.4)	-	0.5 (0.1–1.6)
Immigrants from Maghreb	1.4 (0.8–2.6)	2.5 (1.3–4.7)	1.5 (1.1–2.1)	1.5 (1.2–1.8)
Immigrants from all other African countries	1.2 (0.8–1.9)	2.0 (1.1–3.8)	0.8 (0.5–1.3)	1.0 (0.8–1.3)
Immigrants from Asia	0.1 (0.0–0.2)	0.4 (0.2–1.1)	0.7 (0.4–1.2)	0.6 (0.3–0.9)
Immigrants from European Union 15 countries ^d^	0.5 (0.3–0.9)	1.1 (0.6–2.0)	0.8 (0.5–1.3)	0.8 (0.6–1.1)
Immigrants from all other European countries	2.6 (0.8–8.3)	0.9 (0.2–3.3)	0.9 (0.4–2.3)	1.4 (0.7–2.7)
Immigrants from other countries	0.6 (0.2–1.6)	1.5 (0.6–4.0)	1.2 (0.6–2.5)	1.1 (0.7–1.8)

^a^ HCW: Healthcare workers. ^b^ The association of occupational group and income level is adjusted for gender, age, and geographical origin, while other associations are adjusted for gender, age, geographical origin, and education level. ^c^ Interpretation: compared with men, women are 5.4 times more likely to work as hospital HCWs than as non-key workers. ^d^ The OECD definition of European Union 15 countries (EU15): https://stats.oecd.org/glossary/detail.asp?ID=6805 (accessed on 14 June 2022).

**Table 3 ijerph-19-07741-t003:** Estimated adjusted prevalence ratio (PR) of exposure to the three work-related exposure factors in each sub-group of key workers, compared to non-key workers (stratified models).

	Exposure to Infectious Agents		Face-to-Face Contact with the Public		Working with Colleagues	
	% Exposed	Adjusted PR ^b^ (95% CI)	% Exposed	Adjusted PR ^b^ (95% CI)	% Exposed	Adjusted PR ^b^ (95% CI)
**Non-key workers**	25.3 (24.8–25.9)	Ref	58.4 (57.8–59.0)	Ref	41.4 (40.8–42.0)	Ref
**Hospital HCW ^a^**	89.9 (88.5–91.4)	3.5 (3.3–3.7)	95.0 (94.0–96.0)	1.5 (1.5–1.6)	62.9 (60.7–65.2)	1.6 (1.5–1.7)
**Non-Hospital HCW ^a^**	75.0 (72.7–77.2)	2.9 (2.7–3.1)	91.4 (89.9–92.9)	1.5 (1.4–1.6)	19.5 (17.4–21.6)	0.5 (0.4–0.6)
**Essential Non HCW ^a^**	34.7 (33.3–36.2)	1.4 (1.2–1.5)	72.2 (70.8–73.6)	1.2 (1.2–1.3)	37.7 (36.2–39.2)	0.9 (0.8–1.0)
**All key workers**	55.9 (54.7–57.0)	2.1 (2.0–2.3)	81.4 (80.5–82.3)	1.4 (1.3–1.4)	40.3 (39.2–41.5)	1.0 (0.9–1.1)
**All active population**	31.9 (31.4–32.4)		63.3 (62.8–63.8)		41.2 (40.6–41.7)	

^a^ HCW: Healthcare workers. ^b^ Adjusted for gender, age, and geographical origin.

## Data Availability

Restrictions apply to the availability of these data. The National Archive of Data from Official Statistics (ADISP) authorized access to the anonymized data for secondary analysis (request No 25798). The demand for excess the data could be made from: http://www.progedo-adisp.fr/enquetes/XML/lil.php?lil=lil-1459 (accessed on 23 June 2021).

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
