# Peer review of "Socio-Demographic Composition and Potential Occupational Exposure to SARS-CoV2 under Routine Working Conditions among Key Workers in France"

_ijerph, 2022, doi:10.3390/ijerph19137741_

Round 1
Reviewer 1 Report
This is a useful paper essentially describing the distribution of potentially important sociodemographic variables across key and non-key workers, with an emphasis also on health care front line workers. The collection of this data in and of itself, being essentially cross-sectional and not related to subsequent outcomes, can only be judged as descriptive and baseline. In that sense these data will likely be more useful as a longer followup ensues, relating illness, morbiditiy and mortality rates to the potential inequities pointed out here.
Author Response
Dear reviewer, thank you very much for reviewing our manuscript and providing us with your feedback. Indeed, it was our intention to provide a basic description of socio-professional inequalities in exposure to SARS-CoV-2 that could help further research to investigate the relationship between occupational exposure to the virus and infection, morbidity and mortality.
This article is written as part of a PhD thesis with three main objectives:
Objective 1- To describe and characterize socio-occupational inequalities in the work-related exposure factors to COVID-19 and their eventual change throughout the epidemic
Objective 2- To compare the risk of infection by COVID-19 between different socio-occupational groups, taking into account work-related exposures and housing conditions
Objective 3- Focusing on COVID-19 infected cases, to compare disease severity and fatality between different socio-occupational groups, taking into account co-morbidities and lifestyle habits, known to affect prognosis and sequelae
This work will therefore be completed by studying the dynamics of occupational exposure to COVID-19 and the relationship between occupational exposure factors and COVID-19 infection, hospitalization and mortality using real time epidemic data.
Reviewer 2 Report
This is a relevant and well-conducted study. I have two main points that are important to be addressed:
Line 87, page 2- The study needs a clear objective. This objective is not clear in the main text and also in the abstract. Information about data collection needs to be in the methods section and not in the Introduction.
Instead of subheading “Context”, I recommend including this as part of the Introduction in a paragraph.
2. Materials and Methods, please include the type of Study Design.
Reviewer 3 Report
The present study focuses on sizing the potential risks of SARS-CoV-2 infection in key workers during the first confinement in France. It also puts a scenario little visualized for the importance of the problem, such as the disparities between key workers regarding their social origin, geographic origin, and exposure to SARS-CoV-2.
The study presented is cross-sectional and is adequately planned and conducted. In addition, the authors make an outstanding introduction to make the reader aware of the problem addressed and appropriately address the other sections of the manuscript.
Below, I briefly set out observations (all of which are minor).
Line 92. Because the word race has multiple meanings and because, in biological terms, there are no races in human beings (10.1002/ajpa.23882), I suggest that the word be changed to ethnicity.
Lines 103-108. The sentence is too broad and complicated to follow; I recommend restructuring it (shorten it or split it ) to give a more direct message to the audience.
The authors need to standardize the terms of sex and gender; sometimes, they use them interchangeably.
It is confusing that the authors cite a source in their tables; if the manuscript is original, why do they cite sources, I understand that they were based on CT-2019, but when citing a source, it gives the impression that the source is not their own and therefore the data are not original, I recommend modifying, deleting or being more specific what they mean by "source."
The last four immigrant classifications must start with a capital letter.
It would be significant, during the discussion, to briefly address aspects related to the marginalization in which some of the less favored key workers in the study currently live, as well as epidemiological issues.
If possible, point out how many deaths there were in this population compared to other populations of France.
Reviewer 4 Report
The paper is an interesting work and also well-written. It aligns with the aim of the IJERPH journal and the special issue. The work targeted to examine the social gradient among key workers during the COVID-19 period in France, which is important and worth to be discussed.
Issues of e.g. the low income and risk of exposure to Covid-19 may also increase the mental health problems (e.g. work pressure) are also among these less-visible workers. This further affects their safety actions at work which may further affect their exposure to SARS-CoV-2. This may be discussed in the Discussion Section.
I recommend the journal accept the work.
Reviewer 5 Report
I have attached my comments as a document.

Round 2
Reviewer 5 Report
I would like to thank the authors for thoroughly addressing all of the comments I made. I particularly appreciate the thoughtful responses around race/ethnicity and sex/gender. I also appreciate the removal of the Chi-squared assessment.
Two remaining items need addressing prior to publication, in my opinion
1) Please elaborate on the weighting described in the responses to my comments or at least cite the reference for these weighting methods.
2) Right-justify the numbers in the tables. They are easier to read when not centered.
Author Response
Dear reviewer, thank you very much for your second round of comments. Below are the new revisions to the manuscript accordingly.
1- We added the following sentence to line 179 of the manuscript:
“An extensive description of the sampling and weighting methods used in CT periodic surveys is available elsewhere [26].”
2- The following reference is added:
“A. Mauroux, S. Amira, C. Mette, C. Beswick, and C. Dennevault, “L’enquête Conditions de travail-risques psychosociaux 2016 : apurement et redressement.” DARES, 2020. [Online]. Available: https://dares.travail-emploi.gouv.fr/sites/default/files/pdf/dares_document-etudes_condition_travail_risques_psychosociaux_redressement_apurement_2016.pdf”
3- Numbers in the tables are right-justifies.